# Co-Expression Network and Integrative Analysis of Metabolome and Transcriptome Uncovers Biological Pathways for Fertility in Beef Heifers

**DOI:** 10.3390/metabo12080708

**Published:** 2022-07-29

**Authors:** Priyanka Banerjee, Soren P. Rodning, Wellison J. S. Diniz, Paul W. Dyce

**Affiliations:** Department of Animal Sciences, Auburn University, Auburn, AL 36849, USA; pzb0035@auburn.edu (P.B.); rodnisp@auburn.edu (S.P.R.); wzd0027@auburn.edu (W.J.S.D.)

**Keywords:** beef heifer, data integration, fertility, metabolome, transcriptome

## Abstract

Reproductive failure remains a significant challenge to the beef industry. The omics technologies have provided opportunities to improve reproductive efficiency. We used a multistaged analysis from blood profiles to integrate metabolome (plasma) and transcriptome (peripheral white blood cells) in beef heifers. We used untargeted metabolomics and RNA-Seq paired data from six AI-pregnant (AI-P) and six nonpregnant (NP) Angus-Simmental crossbred heifers at artificial insemination (AI). Based on network co-expression analysis, we identified 17 and 37 hub genes in the AI-P and NP groups, respectively. Further, we identified *TGM2*, *TMEM51*, *TAC3*, *NDRG4*, and *PDGFB* as more connected in the NP heifers’ network. The NP gene network showed a connectivity gain due to the rewiring of major regulators. The metabolomic analysis identified 18 and 15 hub metabolites in the AI-P and NP networks. Tryptophan and allantoic acid exhibited a connectivity gain in the NP and AI-P networks, respectively. The gene–metabolite integration identified tocopherol-a as positively correlated with ENSBTAG00000009943 in the AI-P group. Conversely, tocopherol-a was negatively correlated in the NP group with *EXOSC2*, *TRNAUIAP*, and *SNX12*. In the NP group, α-ketoglutarate-*SMG8* and putrescine-*HSD17B13* were positively correlated, whereas a-ketoglutarate-*ALAS2* and tryptophan-*MTMR1* were negatively correlated. These multiple interactions identified novel targets and pathways underlying fertility in bovines.

## 1. Introduction

Infertility has a negative impact on beef cattle production and the profit of the livestock industry. Despite technological advancements and improvements in management practices [1], reproductive failure remains a concern for cow-calf production due to the amount of capital invested [2]. Additionally, reproductive failure is one of the main reasons for culling, which reduces the longevity of the herd and affects the total profit of the producers [3]. Reproductive success relies on many factors, including breeding strategy, nutritional and health management, and genetic background [4]. Therefore, best management practices in heifer development often increase reproductive success [2].

Predominantly, producers have used traits, such as body condition score [5,6,7], reproductive tract score [8,9], and pelvic measurements [10,11], as tools to identify heifers with a high potential for fertility [5,6,7,8,9,10,11]. However, many heifers deemed reproductively mature even after the phenotypic assessments, fail to conceive [12]. Thus, new methods or tools are still required to identify and select animals with high fertility potential. Among the options, omics technologies have provided opportunities to understand the genetic mechanisms underlying fertility-related traits in cattle. Furthermore, the implementation of genomic evaluation and selection has increased the rate of genetic improvement [13]. Likewise, transcriptomics and metabolomics have provided promising results to assess fertility and identify potential biomarkers and gene signatures.

A plethora of transcriptomic studies from embryo [14,15], endometrium [16,17,18,19,20], uterine tissue [21], and peripheral white blood cells [22,23] have identified candidate genes for bovine fertility. Moorey et al. identified *RPL39*, *SMIM26*, *N6AMT1*, *LONRF3*, and *GATA3* as differentially expressed in peripheral white blood cells associated with pregnancy outcomes in beef heifers [22]. Mazzoni et al. identified regulators, such as *IGF1*, *AGT*, *TNF*, *PI3K*, *FGF2*, *TGF-β1*, and related biological mechanisms important for pregnancy establishment in endometrial transcriptome [18]. Similarly, Geary et al. identified *IGJ*, *IGLL1-2*, *CLCA1*, *PLA2G2F*, and *SHISA6* as differentially expressed in the endometrial biopsies of subfertile and fertile heifers [24]. Likewise, metabolomic profiling of blood [25,26,27] and follicular fluid [28,29] shed light on fertility-related metabolites. Gomez et al. identified pregnancy-specific metabolomic biomarkers, such as 2-oxoglutaric acid, 2-hydroxybutyric acid, dimethylamine, and ketoleucine, in the blood plasma of recipient beef cows [30]. Despite the insights into the novel genes or metabolites reported in these studies, these approaches rely solely on data from a single biological layer. In addition, differential expression analysis disregards the multiple interactions between genes or metabolites and their coordinated expression pattern across samples. Thus, a holistic approach that exploits the genomic layers and their interactions is still required to improve our understanding of the genetic architecture of fertility-related traits.

While the abovementioned studies provide insights into the molecular basis of fertility, to the best of our knowledge, there is a paucity of information about the integrated relationship between fertility, gene expression, and metabolite abundance. We hypothesized that fertility potential is partially determined by the co-expression of genes and metabolites and their coordinated interaction. Thus, our main objective was to unravel the transcriptomic and metabolomic architecture and the biological pathways underlying heifer fertility. To accomplish this, we integrated peripheral white blood cells transcriptome and plasma metabolome profiles using a network approach to unveil functional candidate genes and metabolites involved with heifer fertility. Additionally, we shed light on fertility-related pathways derived from the gene–metabolite pairs.

## 2. Results

### 2.1. Transcriptome and Metabolome Profiles

We used the RNA-Seq expression profile from peripheral white blood cells and the plasma metabolome to ascertain the putative mechanisms underlying fertility differences between AI-P and NP; six animals were in each group of beef heifers. A schematic representation of the study design and analysis steps is given in Figure 1.

After quality control (see Section 4.2), we maintained 10,832 genes out of 24,616 from 12 heifers for differential expression analysis. Based on a combined approach using edgeR and DESeq2 (see Section 4), we identified 38 differentially expressed genes (DEGs, *p*-value ≤ 0.05 and absolute (log2 fold change ≥ 0.5)) (Figure 2, Appendix A). Among the DEGs, 29 genes were upregulated and 9 were downregulated in the NP group. Upregulated genes (top five in the list) included *TFF2*, *TAC3*, *CNKSR3*, *ALAS2* and *MYH10*, while the downregulated genes (top five in the list) included *SAXO2*, ENSBTAG00000039555, ENSBTAG00000034871, *ART3*, and ENSBTAG00000048169.

### 2.2. Gene Network and Functional Over-Representation Analyses

To identify genes or metabolites with coordinated expression patterns within the AI-P and NP groups, we used a co-expression network approach based on the partial correlation and information theory (PCIT) algorithm [31]. For each group, two gene networks and two metabolite networks were created.

For the gene network, the correlation of 10,832 genes identified 3,263,960 significantly correlated gene pairs in the AI-P group, while 5,058,326 significant correlations were identified for the NP group. To reduce data dimensionality and retrieve biologically meaningful information, we maintained the significantly correlated pairs with an absolute correlation greater than 0.99 (*p*-value ≤ 0.05) and co-expressed with the DEGs (Appendix A). After filtering, we maintained 320 and 844 significantly correlated gene pairs for the AI-P and NP groups, respectively (corresponding to 340 and 802 unique genes) (Appendix A).

To identify hub genes, we used the degree and betweenness centrality parameters from the Network Analyzer (Appendix A). After filtering, 17 hub genes were identified in the AI-P and 37 hub genes in the NP networks (Appendix A). Among them, *CYTH3*, *SPN*, *TMEM51*, *GRB10*, *FHIP1A*, *TLCD3A*, *TGM2* and *PDGFB* were identified as shared hubs in both the AI-P and NP networks. The *ENSBTAG00000047816*, *ENSBTAG00000015193*, *ENSBTAG00000009943* and *ABI2* genes were unique to the AI-P group and over-represented biological processes, including innate immune response, regulation of actin dynamics for phagocytic cup formation, and signaling by VEGF and RHO GTPases activating WASPs and WAVEs signaling pathways. The *SH3BP4*, *PDGFA*, *ART3*, *TAC3*, *PLEKHF1*, *NDRG4*, *SMG8*, *PCBP1*, *F2RL2*, *DIP2C*, *EXOSC2*, *ZNF16*, *TRNAU1AP*, *TTK*, *ALAS2*, *HSD17B13*, *SNX12*, *WWOX*, *MTMR1* and *TFF2* genes were unique to the NP group. However, no significant biological processes were over-represented for these genes.

To visualize differences in connectivity between the AI-P and NP gene networks, we built a central reference network using DyNet. This network comprised 1095 nodes (genes) and 1164 edges (interactions) (Figure 3a). The nodes were filtered for betweenness centrality to highlight the hub genes from the AI-P and NP networks (Figure 3b).

The differential connectivity was calculated between the groups based on the frequency of the gene degree, followed by a z-score transformation for the 1095 genes. For the z-scores, 97.5 percentile points for standard normal distribution were considered to retrieve the genes with a z-score ± 1.96 (*p*-value ≤ 0.05). The NP group was used as the reference to determine the gain or loss in gene connectivity (Appendix A). Among the 1095 genes, *PLEKHF1*, *TGM2*, *TMEM51*, *TAC3*, *NDRG4* and *PDGFB* exhibited a gain in connectivity in the NP group. Conversely, *CYTH3*, *ENSBTAG00000027962*, *ENSBTAG00000034871* and *ENSBTAG00000047816* genes were more connected in the AI-P group.

The co-expressed genes, including DEGs, hubs, and differentially connected genes, were used for pathway over-representation analysis separately for the AI-P and NP groups. For the AI-P, over-represented pathways were oocyte meiosis, Wnt signaling pathway, glucagon signaling pathway, propanoate metabolism, N-glycan biosynthesis and protein processing in the endoplasmic reticulum. Likewise, in the NP group, over-represented pathways included MAPK signaling pathway, ubiquitin-mediated proteolysis and mitophagy (Appendix A).

### 2.3. Metabolite Network Analyses

Similar to the gene network analysis, we used PCIT to identify co-expressed metabolite pairs. To this end, after quality control, 112 and 116 metabolites were maintained for the AI-P and NP groups, respectively. The PCIT analysis found 6216 significantly correlated metabolite pairs in the AI-P group (*p*-value ≤ 0.05). Similarly, 6670 metabolite pairs were identified in the NP group. Previously, Phillips et al. reported fifteen differentially expressed metabolites (DEMs) [25]. We then filtered these networks based on the 15 DEMs and significantly correlated with an absolute correlation greater than 0.90 (*p*-value ≤ 0.05). Considering this approach, 18 and 15 metabolite pairs for the AI-P and NP groups (corresponding to 23 unique metabolites in each) were maintained for further analysis (Appendix A).

Based on the degree and betweenness centrality topological parameters calculated by NetworkAnalyzer, we found eight and four hub metabolites for the AI-P and NP networks, respectively (Appendix A). Ornithine and cysteine were identified as hubs of both the AI-P and NP networks. Allantoic acid, methionine, putrescine, phenylethylamine, kynurenine and xylitol were unique to the AI-P group. Likewise, tryptophan and glutamine were unique to the NP network. The central reference metabolite network was constructed with DyNet for the AI-P and NP and comprised 37 nodes and 33 edges (Figure 4).

Next, we identified differentially connected metabolites between the AI-P and the NP groups. The differential connectivity was calculated between the groups based on the frequency of the metabolite degree for the 37 metabolites. Metabolites with z-scores ± 1.96 were significantly differentially connected (*p*-value ≤ 0.05). The gain or loss connectivity was calculated based on the NP group (Appendix A). Among 37 metabolites, tryptophan and allantoic acid were more connected in the NP and AI-P networks, respectively.

### 2.4. Gene–Metabolite Interaction Network

A combined PCIT analysis for 122 metabolites and 10,832 genes resulted in 3,280,267 and 5,075,367 significant co-expressed pairs, including gene–gene, metabolite–metabolite and gene–metabolites pairs, which were retrieved in the AI-P and NP groups, respectively. Only the significant gene–metabolite correlations were filtered based on the unique targets (see Section 4) from the AI-P and NP groups. To identify those targets, we considered the hub genes and metabolites (Appendix A) and the significantly differentially connected genes and metabolites (Appendix A) for each group. Thus, we retrieved 29 and 43 unique genes and metabolites from the AI-P and NP groups, respectively. Among these, 21 genes and metabolites (targets) were shared between both groups, eight were exclusive to the AI-P group and 22 to the NP group (Figure 5). The unique targets were then used as features for selecting gene–metabolite pairs with a correlation greater than 0.75 (*p*-value ≤ 0.05). Based on that, we identified 1161 gene–metabolite pairs in the AI-P network and 155 in the NP networks (Figure 6, Appendix A). These pairs were then used for functional over-representation analysis in MetaboAnalyst (Appendix A). In the AI-P group, the gene–metabolite pairs were over-represented by oocyte meiosis, progesterone-mediated oocyte maturation, and the PI3-Akt signaling pathway. In contrast, glyoxylate and dicarboxylate metabolism, alanine, aspartate and glutamate metabolism, and glucagon metabolism were over-represented in the NP group.

To retrieve the gene–metabolite pairs based on the Integration of Linear model (IntLIM) approach, we utilized 1098 genes and 40 metabolites that were differentially expressed (DEGs and DEMs) or targets (co-expressed, hub genes, and differentially connected) for the AI-P and NP groups. After filtering for 5% variation, 1043 genes and 40 metabolites were maintained for data integration. This analysis resulted in 3177 gene–metabolite pairs (*p*-value ≤ 0.05) (Appendix A). Next, we maintained 714 pairs (Appendix A) by filtering with the unique exclusive targets as given in Figure 5.

Lastly, we overlapped the results from the PCIT and IntLIM to explore those gene–metabolite pairs that were identified in both approaches. These pairs included tocopherol-α, which was positively correlated in the AI-P group with ENSBTAG00000009943 and negatively correlated in the NP group with *EXOSC2*, *TRNAUIAP* and *SNX12*. In the NP group, the α-ketoglutarate-*SMG8* and putrescine-*HSD17B13* pairs were positively correlated, whereas α-ketoglutarate-*ALAS2* and tryptophan-*MTMR1* were negatively correlated.

## 3. Discussion

Currently, beef heifers are selected based on phenotypic assessments. However, a proportion of heifers that pass phenotypically remain NP after the breeding season. Understanding the genetic basis involved in complex traits, such as fertility, will be advantageous in efficiently selecting beef heifers and benefit cow-calf production. The phenotypic outcome of genomic and transcriptomic influences on the molecular state of the cell is reflected by the metabolome [32]. Therefore, in the context of reproductive biology, we focused on a multilayered approach to how regulation of upstream gene expression patterns exert changes on downstream metabolic indicators in the AI-P and NP beef heifers. In our study, we used the RNA-Seq approach to measure the expression of genes and untargeted metabolomics to analyze the metabolite concentration in the blood. Although fertility involves the regulation of multiple tissue organs, we focused on transcriptome and metabolome profiles in the blood because of two reasons: (i) The measurement of a specific product in the blood would be a noninvasive and valuable biomarker for predicting fertility in beef heifers, and (ii) circulating free-RNA and metabolites in the blood provides a potential window into health, phenotype, and developmental status of the organs in mammals [33,34,35]. Studies have reported blood biomarkers that predict infertility or pregnancy outcomes in humans [36,37] and cattle [30]. Furthermore, changes in metabolism affect the transport of nutrients from the blood to the uterine fluid altering fertility [30]. Considering all these factors, we used peripheral white blood cells and plasma to identify novel targets for fertility in beef heifers.

To systematically identify all molecules and their interactions exhibited in the blood for fertility, we used the gene and metabolite co-expression network approach. This approach identified the correlations and recognized the genes and metabolites playing a role in fertility. To accomplish this, we first identified the DEGs and retrieved the DEMs from our previous study [25]. Next, we built co-expression networks and used the DEGs and DEMs as features to prioritize candidate genes and metabolites. We identified the hubs and differentially connected genes and metabolites with significant biological roles, followed by an integrated analysis of the gene–metabolite pairs underlying fertility. Additionally, we identified a gain in gene connectivity in the network from the NP group compared to the AI-P group. This suggests network rewiring of major regulators modulating target gene expression with response to fertility status.

We identified 38 DEGs and 15 DEMs from blood plasma between the AI-P and NP beef heifers. Among the DEGs, *TAC3* (tachykinin precursor 3) was identified as upregulated and a hub with a network connectivity gain in the NP group. The *TAC3* gene encodes a neuropeptide essential to regulating human reproductive function. Supporting human reproductive biology, *TAC3* has been identified in the bloodstream of patients with central pubertal disorders and hypogonadotropic hypogonadism [38,39]. The protein encoded by *TAC3* modulates gonadotropin-releasing hormone (GnRH), which is responsible for the secretion of follicle-stimulating hormone (FSH) and luteinizing hormone (LH) [40]. However, *TAC3* requires the presence of sex steroids to increase FSH and LH secretion [41]. Thus, the upregulation of *TAC3* identified in blood, and yet the heifers not becoming pregnant may suggest an alteration in the hormonal and sex-steroid levels. FSH and LH signaling also regulate several pathways in ovarian granulosa cells, such as the mitogen-activated protein kinase (MAPK) pathway [42]. The MAPK pathway consists of protein kinases that regulate cell proliferation, inflammatory response, development, differentiation, and apoptosis [42]. In our study, the MAPK pathway was significantly over-represented in the NP group. Among the associated genes acting on the MAPK pathway, we identified *PDGFB* (platelet-derived growth factor) as significantly upregulated and differentially connected with a connectivity gain in the NP group. *PDGFB*, identified as a constituent of blood serum and platelets [43], modulates the primordial to primary follicle transition [44], which is essential for female fertility [45,46]. Studies have reported that *PDGFB* signaling promotes epithelial–mesenchymal transition (EMT), which is essential for embryogenesis [47]. EMT regulates critical processes during embryo development, and in the absence of EMT, development cannot proceed past the blastula stage [48]. The MAPK pathway allows cells to interpret external signals, specifically during EMT [49]. Furthermore, the MAPK pathway regulates extracellular signal-regulated kinase (ERK) proteins involved in the primary regulation of GnRH to stimulate the secretion of estradiol and progesterone [42]. In our study, *PDGFB* is negatively and differentially correlated with the hydrocinnamic acid in the NP group. Hydrocinnamic acid is an important class of phenolic acid, having antioxidant and anti-inflammatory properties [50]. Phenolic acids, also called polyphenols, have been associated with fertility, development, fetal health, and the bioavailability of nutrients [51]. A previous study reported that polyphenols inhibited the MAPK signaling pathway and mediated oxidative stress and inflammation [52]. Therefore, these findings suggest that the over-expression of the *PDGFB* gene and the downregulation of hydrocinnamic acid in the NP group triggered the MAPK signaling pathway, likely regulating the inflammatory response. Controlled inflammation is necessary for host defense, while an uncontrolled inflammatory response may affect fertility [53]. However, further validation of *PDGFB* and hydrocinnamic acid in reproductive cells or tissue types could highlight precise downstream mechanisms underlying fertility.

We identified the over-represented oocyte meiosis and Wnt signaling pathways in the AI-P group. The genes included *ANAPC13*, *CAMK2G*, *CCNE2*, *PPP2R1B*, *PPP2R5A* and *PPP3CA*. The *CAMK2G* gene encodes a calcium/calmodulin-dependent protein kinase involved with GnRH release [54]. *CAMK2G* was positively correlated with *TGM2* (Transglutaminase 2) as given through PCIT in our study. *TGM2* was upregulated and identified as a hub in the NP group. This result was further supported as *TGM2* was significantly upregulated in uterine fluid collected from infertile women [55]. Furthermore, *TGM2* levels were higher in the blood of patients with endometrial cancer [56]. *TGM2* activates the β-catenin (Wnt) pathway, which impairs ovulation [57]. Balanced Wnt signaling is essential for the proper development of reproductive tract organs and fertility [58]. A previous study suggested overactivation of Wnt signaling has no effect on folliculogenesis but causes fertility failure due to abnormal fetal development [58]. Although the Wnt signaling pathway was over-represented in the AI-P group, differential regulation of co-expressed genes may regulate infertility in the NP group.

The PI3-Akt signaling pathway was over-represented by significant gene–metabolite pairs in the AI-P group. PI3-Akt signaling pathway plays an essential role in cell proliferation, apoptosis, metabolism, regulation of dormancy, and activation of mammalian primordial follicles [59]. Decreased PI3-Akt promotes Bax (a pro-apoptosis protein) translocation to the mitochondria and subsequent release of cytochrome c triggering apoptosis through the caspase pathway [59]. Higher apoptosis of granulosa cells has been associated with empty follicles and fewer or poor-quality oocytes and embryos in humans [60]. Furthermore, emerging evidence indicates the interrelationship of the PI3-Akt signaling pathway with pregnancy outcomes [61]. Akt deficient mice exhibited fewer mature follicles, reducing fertility [62]. This indicates the potential role of PI3-Akt in the AI-P group. This was further supported by one of the genes, *NDRG4* (N-Myc downstream-regulated gene 4), downregulated in the AI-P group. *NDRG4* has a crucial role in transferring its repression effect on PI3-Akt activity [63]. Studies in humans reported a negative feedback loop between *NDRG4* activation and consequent PI3-Akt suppression [63], which could explain the role of NDRG4 in infertility via PI3-Akt suppression. Apart from being upregulated, *NDRG4* was identified as a hub with a network connectivity gain in the NP group. Previous studies reported the *NDRG4* expression in other reproductive tissue types [64], but its role in bovine fertility has not been reported yet. These findings suggest that *NDRG4* has a potential role in regulating fertility and could be further explored.

Insulin-like growth factors modulate pathways, such as MAPK and PI3-Akt, by regulating gamete development, quality, and implantation [65]. Our study identified *GRB10* (growth factor receptor bound protein 10) that was upregulated and differentially connected (loss in connectivity) in the NP group. *GRB10* interacts negatively with insulin and insulin-like growth factor receptors [66,67,68]. Paulo et al. reported the involvement of the *GRB10* gene during oocyte maturation in the bovine cumulus–oocyte complex [69]. The association of single nucleotide polymorphism in *GRB10* with superovulation traits in cattle suggested the recovery of many embryos from heterozygous *GRB10* cattle [70]. These studies suggested the role of *GRB10* in regulating fertility, but a comprehensive understanding of bovine fertility requires further investigation.

The network analysis identified *ART3*, *MTMR1*, *ALAS2*, *TFF2*, *SMG8*, *PLEKHF1* and *EXOSC2* as unique hub genes in the NP group. *ART3* (ADP-Ribosyltransferase-3) was downregulated and positively correlated with ornithine, putrescine and alpha-ketoglutarate in the NP group. As a representative arginine-specific ART-ribosyltransferase, *ART3* has been reported to play a critical role in cell division, DNA repair, and inflammatory response [71]. Arginine and ornithine are precursors of nitric oxide and polyamines, respectively. In mammals, the ovaries produce ornithine decarboxylase (ODC) and putrescine during ovulation [72]. In previous studies, aged mice with reduced fertility exhibited low levels of ODC in ovaries; however, that did not improve after diet supplementation [72]. Interestingly, putrescine supplementation during in vitro maturation of aged mouse oocytes improved the quality of blastocysts [73]. These findings leave open questions about *ART3* and its correlated metabolites (ornithine and putrescine) in the NP group.

Tryptophan was identified as differentially connected with a connectivity gain in the NP group. Tryptophan was negatively correlated with *MTMR1* (myotubularin-related protein 1), a hub gene in the NP network. Tryptophan levels are influenced by dietary and hormonal factors [74]. Free tryptophan (not bound to albumin) is the protein carrier in plasma that is available to be taken by tissues and organs [75]. The previous literature shows that, besides fulfilling the demands of maternal proteins, tryptophan is required for fetal growth and development [76]. The need for tryptophan varies as the pregnancy progresses. In early and mid-pregnancy, increased maternal tryptophan availability helps meet the demand for protein synthesis and fetal development in humans [76]. Furthermore, tryptophan is reported as an antioxidant with a suppressive activity against cytochrome P-450-dependent lipid peroxidation in oxidative stress [77]. Additionally, levels of tryptophan were lower in women suffering from preeclampsia [77]. This evidence suggests that high levels of tryptophan are required during the initial phases of pregnancy. Downregulation of tryptophan could be one of the possible reasons for the infertility of beef heifers in our study. Evidence regarding the antioxidant effects of tryptophan has been reported, but the mechanism of action regulating fertility in bovines has not been reported yet [77].

The genes, metabolites, and their interactions identified in the present study over-represented some pathways playing a role in fertility. Some of the genes and metabolites were supported by the previous literature; however, some other targets identified were novel and warrant further detailed studies to evaluate the repeatability of our results in a larger cohort. As the heifers undergo variations during the estrus cycle and pregnancy, it is obvious to expect alterations in genes and metabolite profiles at a particular time point. This means that strategies for applying this information in real-time farming appear to be more complex and need a better understanding of the changes in genes, metabolites, and their interactions at different time points. Validation of these results in more samples and different time points would help to establish a framework for future fertility prediction using gene and metabolite biomarker profiles that could be practical for on-farm use and improve reproductive efficiency. Furthermore, validation of these genes and metabolites in the reproductive tissues and cells would help to identify the therapeutic targets for fertility.

## 4. Materials and Methods

### 4.1. Data Collection, Gene Expression, and Metabolite Profile

The RNA-Seq and metabolomic datasets previously generated by our group and publicly available were used in the current study. The experimental design, sample collection, and data generation were published elsewhere [23,25]. Briefly, blood samples from Angus-Simmental crossbred heifers were collected at AI and used for transcriptomic profiling of peripheral white blood cells and plasma metabolites.

Raw gene counts of 12 samples classified as AI-P or NP with six animals in each group from “station B” were downloaded from the GEO database (accession number GSE103628) [23]. RNA-Seq quality control and mapping were reported by Dickinson et al. [23]. From the same samples, blood plasma untargeted metabolomic profile was performed on gas chromatography (GC) coupled to time-of-flight mass spectrometry (TOF-MS) (GC-TOF-MS—Agilent 6890 GC). Analytical details were described by Phillips et al. [25].

### 4.2. Gene and Metabolite Differential Expression Analyses

Prior to statistical analysis, the read count data were transformed to counts per million (CPM) using edgeR v3.28.1 [78]. For the quality control, raw counts were filtered out based on the following criteria: (i) transcripts with zero counts (unexpressed); (ii) transcripts with less than 1 CPM on average (very lowly expressed); and (iii) transcripts that were not present in at least 50% of the samples (rarely expressed) [79].

We used the following two methods for the differential expression analysis: edgeR and DESeq2. This was performed to increase the robustness of our analysis and capture the biological variation. Thus, DEGs identified by both tools with a *p*-value ≤ 0.05 and absolute (log2 fold change) ≥ 0.5 were considered significant. The NP heifers were used as the reference group, and DEGs were classified as up- or downregulated based on the log2 fold change direction. The upregulated and downregulated genes were visualized using a volcano plot constructed using the R-package EnhancedVolcano v1.8.0 [80]. The fold change (FC) estimates by both the packages were similar. Thus, the FC output from DESeq2 was used.

For metabolomics, raw data files were preprocessed and quantification was reported as peak height using unique ion as default [25]. For each group of metabolites, only those with a relative standard deviation ≤0.15 were used based on the raw counts. Since this study focuses on the interactions between metabolites, the data were log normalized to unfold the differences in the AI-P and NP groups [35,81,82]. Based on this dataset, we retrieved 15 DEMs [25]. The DEMs and DEGs were then used as features for network filtering, as detailed below.

### 4.3. Gene and Metabolite Co-Expression Networks

The partial correlation and information theory (PCIT) approach was used to identify expression patterns and relationships within each regulatory layer (transcriptome and metabolome). PCIT reports the significantly correlated pairs after comparing all possible trios of genes or metabolites [31]. Gene co-expression networks were created considering 10,832 genes maintained after quality control (see methods in Section 4.2). Based on the *normTransform* function from DESeq2, read counts were normalized and used as input for PCIT analysis. For the metabolite networks, 122 metabolites were used after data normalization (see methods in Section 4.2).

Gene and metabolite networks were created separately for the AI-P and NP groups. This approach allowed us to identify changes in the network topology between conditions by comparing the connectivity of genes and metabolites. To reduce data dimensionality, a more stringent correlation value was used to select the gene pairs. Significantly correlated pairs (*p*-value ≤ 0.05) were determined based on partial correlation values greater than absolute (r ≥ 0.99) and absolute (r ≥ 0.9) for genes and metabolites, respectively. Additionally, significant pairs were maintained when a DEG or DEM was co-expressed.

Network topological parameters, including degree and betweenness centrality, were computed by the Cytoscape v3.8.2 [83] plugin Network Analyzer tool [84] and used for identifying hubs. The highly connected hubs were determined using the degree ≥ 1 and betweenness centrality ≥ 0 in each group for genes and metabolites and were referred to as “*hub targets*”. Based on network assumptions, highly connected nodes (genes or metabolites) are central to the network architecture and may be the primary regulators of biologically meaningful pathways.

To visualize changes in nodes and edge rewiring from the AI-P and NP groups, we used the Cytoscape plug-in—DyNet [85]. DyNet provides a central reference network constructed from the co-expressed pairs and highlights the most rewired nodes between the groups. Further, we identified the differentially connected genes or metabolites within each network. To this end, we calculated the differential connectivity (*DK*) index, which is the ratio of the node (gene or metabolite) connectivity to the maximum connectivity for each group [86]. The index is given by the following equation: $DK_{i}=K_{NP\left(i\right)}-K_{AI-P\left(i\right)}$. To determine the significance level, the index was transformed into a z-score (z−score=(DKi−μ)/σ), where μ is the mean *DKi* and σ  is the standard deviation of *DKi*. Significantly differentially connected genes or metabolites (referred to as differentially connected targets) were retrieved when z-score values were between ± 1.96 from the standard deviation (*p*-value ≤ 0.05). The networks were constructed and visualized using Cytoscape v3.8.2.

### 4.4. Transcriptomic and Metabolomic Data Integration

Two approaches were used to unravel the complex relationship between genes and metabolites within the AI-P and NP groups. First, we created a single co-expression network combining the normalized genes (10,832) and metabolites (122) on PCIT to identify significantly correlated gene–metabolite pairs [87]. Only gene–metabolite pairs with absolute correlation values ≥ 0.75 and co-expressed with a hub or differentially connected targets (see methods in Section 4.3) were considered for further analysis.

To get an insight into the specific networks involved with fertility status, the correlated pairs were filtered with the “*unique*” targets. Herein, unique targets refer to the overlapping hub and differentially connected targets between the AI-P and NP groups. The networks with significant gene–metabolite pairs for the AI-P and NP groups were visualized in Cytoscape. Likewise, nodes and edge rewiring within the interaction networks were performed using DyNet (Cytoscape v3.8.2).

The second approach was based on a linear model framework using the IntLIM (Integration of Linear model) R-package [88]. IntLIM integrates the metabolomic and transcriptomic data considering a case-control design. For the present study, the differentially expressed (DEGs and DEMs), co-expressed hubs, and differentially connected targets previously identified from genes and metabolites for the AI-P and NP groups were used for the integration. The genes with the lowest 5% variation were excluded from the study as a quality control step. The linear model for data integration is given below:*m* = *b*_1_ + *b*_2_
*g* + *b*_3_
*p*+ *b*_4_ (*g*:*p*) + *e*
where:*m*: is the log-normalized metabolite abundance;*b*_1_: is the intercept;*b*_2_*g*: is the normalized and adjusted gene-expression level;*b*_3_*p*: is the phenotype (AI-P and NP);*b*_4_ (*g*:*p*): is the interaction between gene expression and phenotype; and*e*: is the residual effect associated with each observation.

The absolute difference in the Spearman correlation was identified between the groups to identify significant gene–metabolite pairs (*p*-value ≤ 0.05). The absolute difference between the groups was estimated as (*rPreg_cor—rNP_cor*), where *rPreg_cor* and *rNP_cor* are the correlation values for the AI-P and NP groups.

### 4.5. Pathway Over-Representation Analysis

The pathway over-representation for DEGs, co-expressed, hub, and differentially connected genes were performed using ClueGO v2.5.4 and ShinyGO webtool. These approaches allowed us to identify the KEGG pathways and significant biological processes underlying differences within each group [89,90]. The redundant terms were clustered with a kappa score of 0.4, and the *Bos taurus* annotation was used as the background in ClueGO. Furthermore, the pathways were selected based on the *p*-value corrected for Bonferroni step down ≤ 0.05.

For the significant gene–metabolite pairs, joint-pathway analysis was performed using MetaboAnalyst 5.0 specific to *Bos taurus*. We used a hypergeometric test for enrichment, betweenness centrality as a topological measure and combined the *p*-values at the pathway level for integration. Pathways with *p*-value ≤ 0.05 (FDR ≤ 0.1) were considered significant.

## 5. Conclusions

The current study focused on identifying putative major regulators associated with fertility. To this end, we investigated the differences in the gene or metabolite expression and co-expression in beef heifers. Additionally, an integrated gene–metabolite analysis approach using correlation and linear model unveiled the potential gene–metabolite pairs affecting biological processes related to fertility in beef heifers. This research’s originality lies in multiple approaches used in determining gene–gene, metabolite–metabolite, and gene–metabolite interactions at AI. A detailed understanding of these targets and their underlying pathways in a bigger cohort at AI and at different time points could establish a framework for fertility prediction. More data and validation of these results would help model the complex relation of gene–metabolite profile over time.

## Figures and Tables

**Figure 1 metabolites-12-00708-f001:**
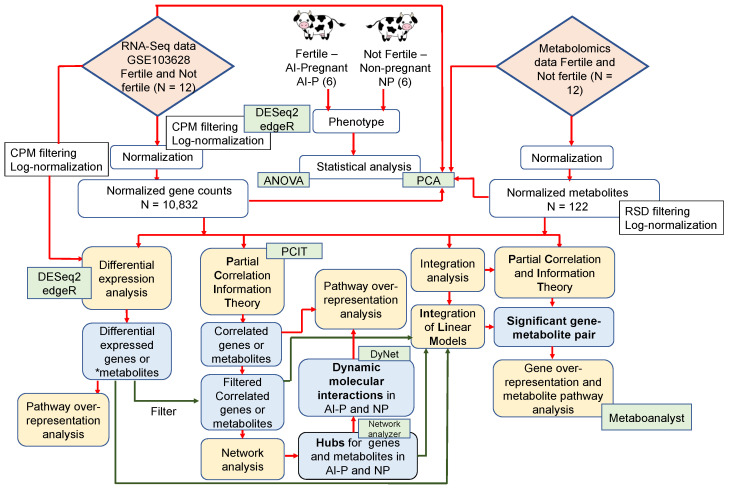
Schematic representation of the study design and analysis steps. The yellow boxes represent the analysis steps; the green boxes represent the software/packages used; and the blue boxes represent the output obtained. The red connectors point to the main analysis steps, while the green connectors are the analysis after filtering. * Differentially expressed metabolites retrieved from Phillips et al. [25].

**Figure 2 metabolites-12-00708-f002:**
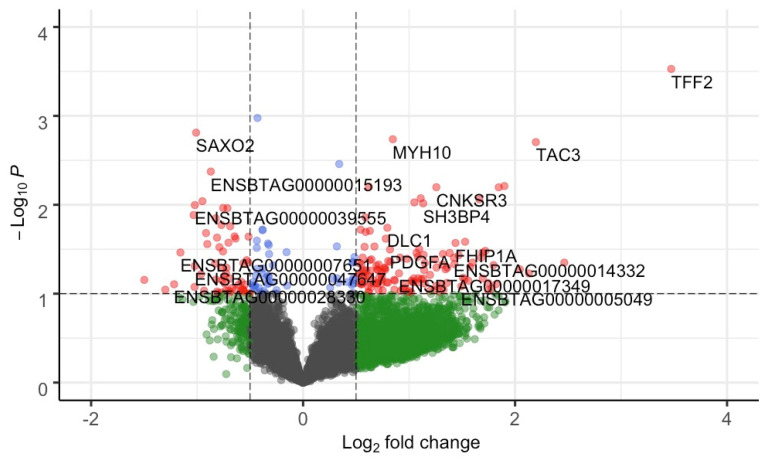
Volcano plot of gene-based differential expression analysis for AI-P (*n* = 6) and NP (*n* = 6) crossbred heifers. Each dot corresponds to a gene. The x-axis represents log2 fold change for the difference in expression in the AI-P and NP groups. The y-axis corresponds to the negative log (base 10) of the *p*-value. Grey dots represent nonsignificant genes that did not cross the threshold of *p*-value or fold-change; green dots represent the genes with absolute (log2 fold change ≥ 0.5); blue dots represent the genes with significant *p*-value; and red dots represent 38 DEGs (*p*-value ≤ 0.05 and absolute (log2 fold change ≥ 0.5)). The genes on the left panel (0 to −2 of log2 fold change) were downregulated, while the genes on the right panel (0 to 4 of log2 fold change) were upregulated.

**Figure 3 metabolites-12-00708-f003:**
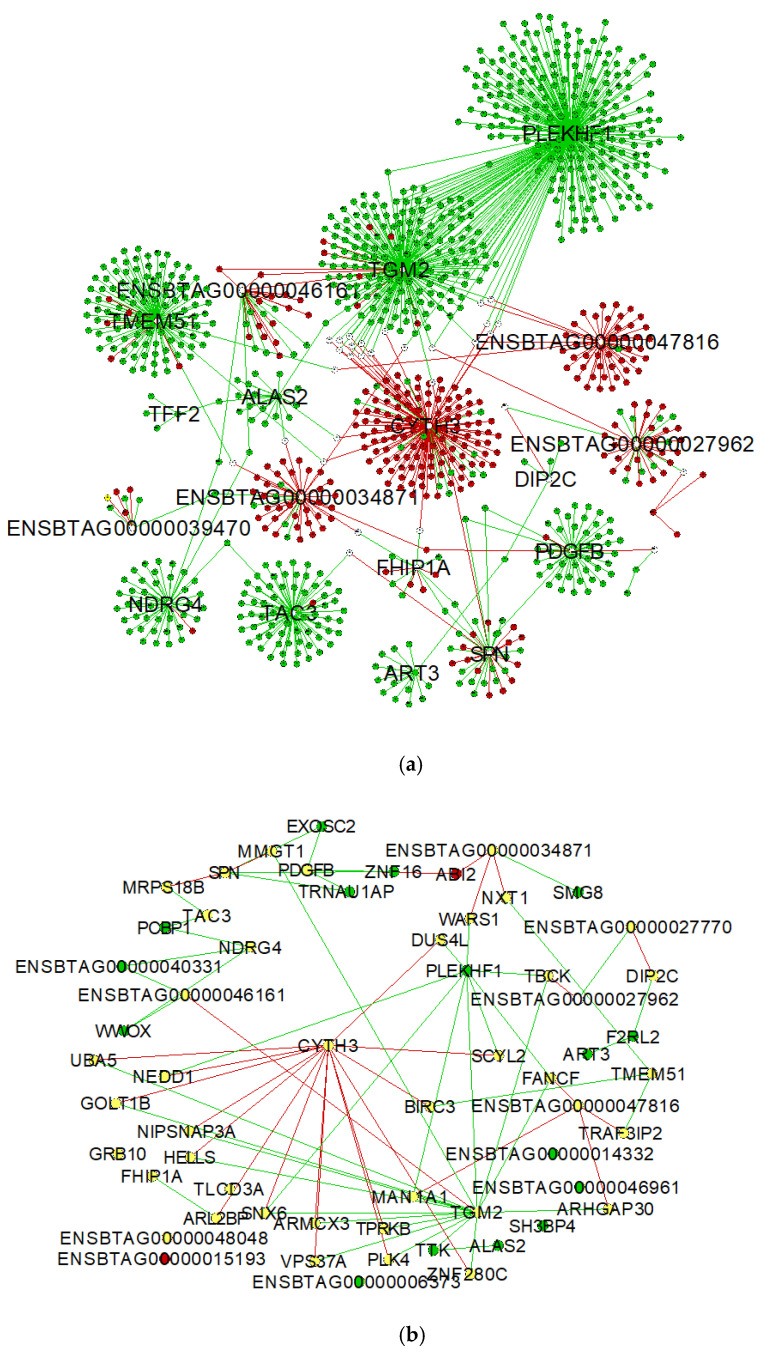
Network comparison based on rewired nodes (genes) of AI-P and NP networks. Central reference network constructed using DyNet in Cytoscape. (**a**) The network comprises 1095 nodes and 1164 edges. The hub targets connected to the network are represented in this network. (**b**) The network was filtered for betweenness centrality to display the hub genes from both groups. The unique nodes are red (AI-P) or green (NP). Shared nodes between AI-P and NP are yellow.

**Figure 4 metabolites-12-00708-f004:**
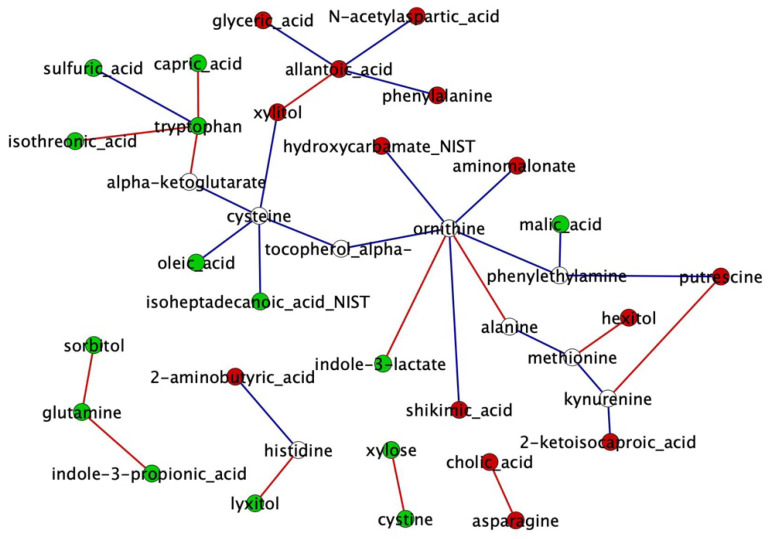
Metabolite network comparison based on rewired nodes (metabolites) from the AI-P and NP groups. A central reference network was constructed using DyNet in Cytoscape. The unique nodes are red (AI-P) or green (NP). Shared nodes between AI-P and NP are white. The edges are red (negative correlation) and blue (positive correlation).

**Figure 5 metabolites-12-00708-f005:**
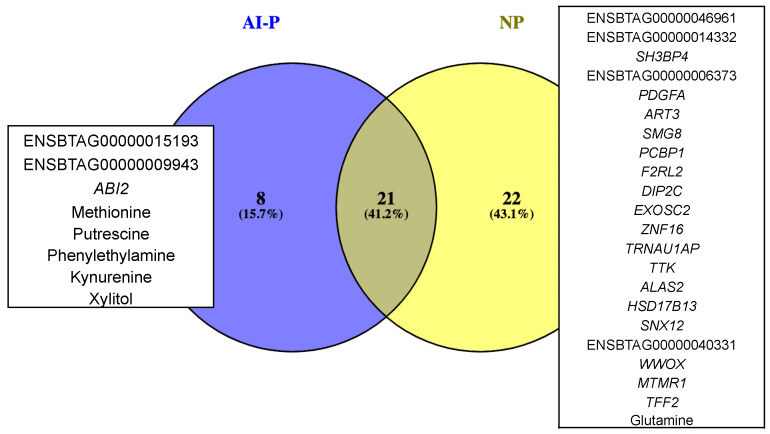
Gene and metabolite targets exclusive for the AI-P and NP groups.

**Figure 6 metabolites-12-00708-f006:**
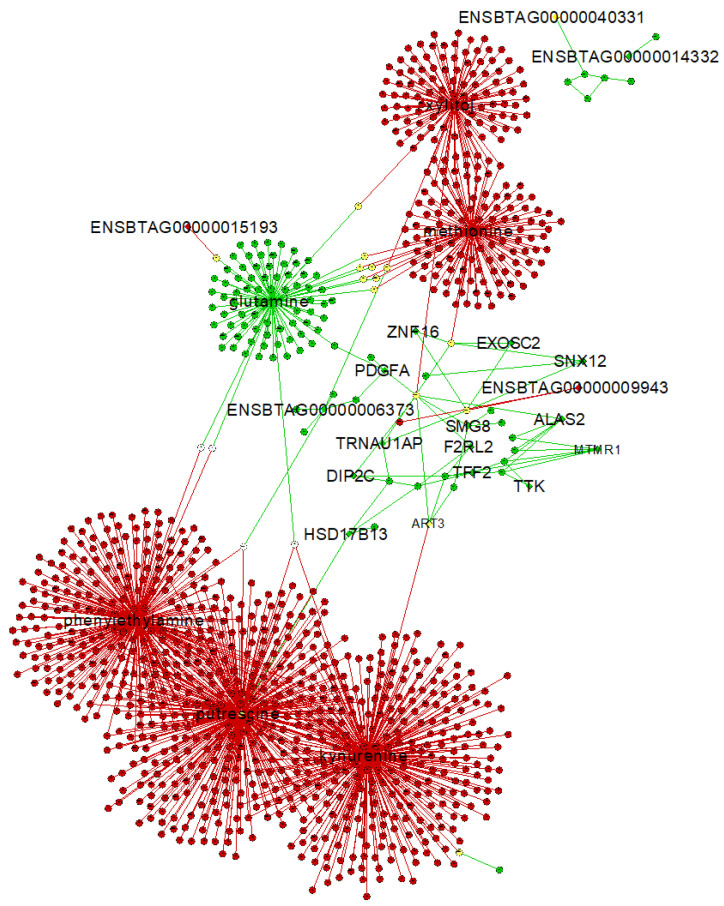
Gene–metabolite interaction network from the AI-P and the NP groups. The unique exclusive targets are highlighted. A central reference network was constructed using DyNet in Cytoscape. The unique nodes are red (AI-P) or green (NP). Shared nodes between AI-P and NP are yellow.

## Data Availability

The transcriptomic data is available in the GEO database (accession number GSE103628). The metabolites data analyzed are available from the corresponding author on reasonable request.

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
