# Peer review of "Co-Expression Network and Integrative Analysis of Metabolome and Transcriptome Uncovers Biological Pathways for Fertility in Beef Heifers"

_metabolites, 2022, doi:10.3390/metabo12080708_

Round 1

Reviewer 1 Report

      Reviewer’s Report

Title

:

Co-expression Network and Integrative Analysis of Metabolome and Transcriptome Uncovers Biological Pathways for Fertility in Beef Heifers

Recommendation

:

Revise - Minor

In my opinion, this article is interesting and well-written, which aims to unravel the transcriptomic and metabolomic architecture and the biological pathways underlying heifer fertility. Besides that, appropriate and adequate references to related and previous work are cited and clearly presented in this article. However, I have some comments for the authors to consider in revising their manuscript:

1.     The introduction section should not include the summary of findings (Lines 72-78).

2.     Line 92: What is the reason to keep 10,832 genes out of 24,616?

3.     Line 104: Figure 2 – Pink dots represent 38….. However, I cannot see the pink dots in Figure 2.

4.     Line 266. Please define the abbreviation at first it appears.

5.     What is the limitation of this study?

Author Response

Thank you for the comments and suggestions. We are providing the proper answers for each question below. The edits applied in the manuscript are tracked in the answers below.

  1. The introduction section should not include the summary of findings (Lines 72-78).

We thank the reviewer for the suggestion. Lines 72- 78 highlighting the summary of the findings have been removed.

  1. Line 92: What is the reason to keep 10,832 genes out of 24,616?

This is a quality control step. Filtering low-expression genes is a common practice because it can increase our confidence in discovering differentially expressed genes (DEGs) and reduce the number of multiple tests. The rationale and criteria for the filtering are in the manuscript as follows:

Line 499 - 502: Raw counts were filtered out based on the following criteria: i) transcripts with zero counts (unexpressed); ii) transcripts with less than 1 CPM on average (very lowly expressed); iii) transcripts that were not present in at least 50% of the samples (rarely expressed). After filtering, we retrieved 10,832 genes from 24,616 for differential expression analysis.

The explanation is provided in methodology section 4.2.

  1. Line 104: Figure 2 – Pink dots represent 38….. However, I cannot see the pink dots in Figure 2.

Thank you for pointing this out. We replaced the color to “red dots” in the figure legend.

  1. Line 266. Please define the abbreviation at first it appears.

We have revised the text and placed the abbreviation TAC3 (tachykinin precursor 3) where it first appeared.

  1. What is the limitation of this study?

We have highlighted the limitations of the study at the end of the discussion section (lines 470 - 482).

The limitations include:

  1. The study was conducted on 12 heifers (6 in each group). Some of the genes and metabolites were supported by previous literature; however, some other targets identified were novel and warrant further detailed studies to evaluate the repeatability of our results in a larger cohort. Thus, repeating the study on more samples would further validate the novel genes and metabolites identified in this study.
  2. In this study, we integrated untargeted metabolomics and RNA-Seq data from six pregnant and non-pregnant beef heifers at artificial insemination (AI). As the heifers undergo variations during the estrus cycle and pregnancy, it is obvious to expect alterations in genes and metabolite profiles at a particular time point. This means that strategies for on-farm application of this information appear to be more complex and need a better understanding of the changes in genes, metabolites, and their interactions in a different time point, such as at weaning.

Reviewer 2 Report

Your article exposes the problem of infertility in the world of the meat industry, which is a problem when it comes to achieving success rates in beef production.

With the new molecular and genetic techniques, it is possible, as you indicate, to discover genetic causes involved in reproductive failure. I consider the use of metabolomics and transcriptomics to unmask potential inter-individual variability in the same situation interesting.

You obtain relevant results, different variants that act differently in the face of reproduction, considering on the one hand posive options, compared to other options that will be more infertile. The number of cattle is relatively low and we could obtain more information by expanding that number, but I think it is a good starting point.

I consider that it is a potentially interesting article for the readers of the magazine.

Regarding the form of the article, I consider that some changes could be made that could improve it, for example the existence of many acronyms, without the full name appearing, the name should be given in full, which would make it easier for the reader to understand the article. Article.

In addition, Figure 1. Schematic representation of the study design and analysis steps... should improve the image quality, and the size of the letters.

With all this, I consider that it is an optimal article for publication if a small series of changes are considered.

Author Response

We appreciate your insights on the manuscript and the suggestions to improve it. Yes, we agree that the sample size is small. However, as an exploratory study, it lays the foundation for a deeper investigation using a large cohort. The limitation due to the sample size was included in the manuscript as follows:

Line 470 - “Some of the genes and metabolites were supported by previous literature; however, some other targets identified were novel and warrant further detailed studies to evaluate the repeatability of our results in a larger cohort.”

Line 477 – “Validation of these results in more samples and different time points would help to establish a framework for future fertility prediction using gene and metabolite biomarker profiles that could be practical to on-farm use and improve reproductive efficiency. Furthermore, validation of these genes and metabolites in the reproductive tissues and cells would help to identify the therapeutic targets for fertility.

We have removed some of the acronyms and replaced them with the full names for the ease of the readers.

The font size of Figure 1 with the representation of the study design is now increased and replaced in the main text.

Reviewer 3 Report

In this study, an integrated gene-metabolite analysis approach using correlation and linear model unveiled the potential gene-metabolite pairs affecting biological processes related to fertility in beef heifers. There are more doubts about the selected samples (blood), but the originality of this study lies in multiple approaches used in determining gene-gene, metabolite-metabolite and gene-metabolite interactions at AI. The method of data analysis is worth learning.

Comment 1, lines 42: Is reproductive maturity after failure of phenotypic assessment? Can you explain more?

Comment 2, lines 75: It is recommended to check the use of "bull" again. Three reproductive cycles for male bulls?

Comment 3, lines 76: It is recommended to use abbreviations after the first occurrence.

Comment 4, lines 91, 165, 336, 392, 395: Keep references in the text consistent.

Comment 5, lines 95: Please explain "abs".

Comment 6: The "p" in the ordinate of Figure 2 does not need italics. In addition, log2 (FC) is suggested to be written in the same way as "log2FC" or "log2 fold change".

Comment 7: It is recommended to enlarge the points in the volcano map.

Comment 8, lines 139: It is recommended to beautify the picture and enlarge it moderately.

Comment 9, lines 151, 175301: I suggest "…A, B and C…" instead of "…A, B, and C…".

Comment 10, lines 234, 235: It is recommended to check "beef replacement heifers" and "open" again.

Comment 11, lines 243-253: The purpose of this study was the fertility of beef cattle. The author explained why blood was used as metabolome and transcriptome instead of reproductive organs. However, whether the detection of blood could represent the fertility of beef cattle and how to exclude other interfering factors?

Comment 12, lines 353, 397: Why italics?

Comment 13, lines 499: Please indent the first line to keep the text consistent.

Author Response

Thank you for the comments and suggestions. 

Comment 1, lines 42: Is reproductive maturity after failure of phenotypic assessment? Can you explain more?

This is an interesting question. The phenotypic assessments are performed by the producers as tools to select heifers. For phenotypic assessments, producers use traits such as body condition score, reproductive tract score and pelvic measurements. But it is found that heifers that have been characterized as reproductively mature after these assessments sometimes fail to conceive. Therefore, we conducted this study that it is not just the phenotype of the animal that plays a role; it is the genes and metabolites that are also playing a role in fertility.

Comment 2, lines 75: It is recommended to check the use of "bull" again. Three reproductive cycles for male bulls?

Yes, there could be chances that the heifer is not pregnant via artificial insemination. The success of AI is influenced by several reasons, including artificial insemination timing, insemination frequency, ovarian stimulation protocols, and infertility. Thus, to conclude that the heifer is not getting pregnant because of a fertility issue, the heifer is exposed to a bull for natural breeding. After AI, the heifer is checked for pregnancy; if it is open, they are exposed to a bull. After three natural cycles of natural breeding, the heifer is examined again for pregnancy. This screening is used to categorize the heifer as fertile or not fertile.

Comment 3, lines 76: It is recommended to use abbreviations after the first occurrence.

Thank you for pointing this out. We have modified the text accordingly.

Comment 4, lines 91, 165, 336, 392, 395: Keep references in the text consistent.

We appreciate your keen observation of the inconsistency with the references in the text. The manuscript has been modified, and all the references have been moved to the end of the sentence.

 Comment 5, lines 95: Please explain "abs".

The abbreviation of abs is modified to “absolute”. Herein, absolute refers to the absolute fold change values independent of the sign (negative or positive).

Comment 6: The "p" in the ordinate of Figure 2 does not need italics. In addition, log2 (FC) is suggested to be written in the same way as "log2FC" or "log2 fold change".

The italics have been removed from the p-value < 0.05. log2FC is now replaced by log2 fold change in the manuscript.

Comment 7: It is recommended to enlarge the points in the volcano map.

The points and the label size have been enlarged in the volcano plot.

Comment 8, lines 139: It is recommended to beautify the picture and enlarge it moderately.

Figure 3a has been modified, and the hub targets are now displayed in the network. The font size of both Figures 3a and 3b has been increased.

Comment 9, lines 151, 175,301: I suggest "…A, B and C…" instead of "…A, B, and C…".

Thank you for the suggestion. The manuscript has been modified to the format “A, B and C…” as suggested.

Comment 10, lines 234, 235: It is recommended to check "beef replacement heifers" and "open" again.

Beef replacement heifers have been modified to “beef heifers,” and open has been modified to “non-pregnant”.

Comment 11, lines 243-253: The purpose of this study was the fertility of beef cattle. The author explained why blood was used as metabolome and transcriptome instead of reproductive organs. However, whether the detection of blood could represent the fertility of beef cattle and how to exclude other interfering factors?

We thank the reviewer for the interesting question. We have included an explanation in the discussion section and references as follows:

(Lines 314 - 324): “Although fertility involves the regulation of multiple tissue organs, we focused on transcriptome and metabolome profiles in the blood because of two reasons: i) The measurement of a specific product in the blood would be a non-invasive and valuable biomarker for predicting fertility in beef heifers, and ii) Circulating free-RNA and metabolites in the blood provides a potential window into health, phenotype and developmental status of the organs in mammals. Studies have reported blood biomarkers that predict infertility or pregnancy outcomes in humans and cattle. Furthermore, changes in metabolism affect the transport of nutrients from the blood to the uterine fluid altering fertility. Considering all these factors, we used peripheral white blood cells and plasma to identify novel targets for fertility in beef heifers”.

We agree that several other factors may affect reproduction. Considering the complexity of reproductive-related traits, unfortunately, it is not possible to isolate all the other external factors. However, we have heifers of similar age, genetic background and managed under the same reproductive protocol to minimize external variation. Therefore, we believe that the individual variability measured in this study is due to fertility potential. Our findings through the combination of metabolomics and transcriptomics pointed out known and novel candidate genes and metabolites underlying pregnancy success.

 Comment 12, lines 353, 397: Why italics?

The italics from in vitro and counts per million have been removed.

Comment 13, lines 499: Please indent the first line to keep the text consistent.

Thank you for your keen observation. The first line of the conclusion section has been indented.